# SDP Relaxation with Randomized Rounding for Energy Disaggregation

**Kiarash Shaloudegi**
Imperial College London
k.shaloudegi16@imperial.ac.uk

**András György**
Imperial College London
a.gyorgy@imperial.ac.uk

**Csaba Szepesvári**
University of Alberta
szepesva@ualberta.ca

**Wilsun Xu**
University of Alberta
wxu@ualberta.ca

## Abstract

We develop a scalable, computationally efficient method for the task of energy disaggregation for home appliance monitoring. In this problem the goal is to estimate the energy consumption of each appliance over time based on the total energy-consumption signal of a household. The current state of the art is to model the problem as inference in factorial HMMs, and use quadratic programming to find an approximate solution to the resulting quadratic integer program. Here we take a more principled approach, better suited to integer programming problems, and find an approximate optimum by combining convex semidefinite relaxations randomized rounding, as well as a scalable ADMM method that exploits the special structure of the resulting semidefinite program. Simulation results both in synthetic and real-world datasets demonstrate the superiority of our method.

## 1   Introduction

Energy efficiency is becoming one of the most important issues in our society. Identifying the energy consumption of individual electrical appliances in homes can raise awareness of power consumption and lead to significant saving in utility bills. Detailed feedback about the power consumption of individual appliances helps energy consumers to identify potential areas for energy savings, and increases their willingness to invest in more efficient products. Notifying home owners of accidentally running stoves, ovens, etc., may not only result in savings but also improves safety. Energy disaggregation or non-intrusive load monitoring (NILM) uses data from utility smart meters to separate individual load consumptions (i.e., a load signal) from the total measured power (i.e., the mixture of the signals) in households.

The bulk of the research in NILM has mostly concentrated on applying different data mining and pattern recognition methods to track the footprint of each appliance in total power measurements. Several techniques, such as artificial neural networks (ANN) [Prudenzi, 2002, Chang et al., 2012, Liang et al., 2010], deep neural networks [Kelly and Knottenbelt, 2015], $k$-nearest neighbor ($k$-NN) [Figueiredo et al., 2012, Weiss et al., 2012], sparse coding [Kolter et al., 2010], or ad-hoc heuristic methods [Dong et al., 2012] have been employed. Recent works, rather than turning electrical events into features fed into classifiers, consider the temporal structure of the data[Zia et al., 2011, Kolter and Jaakkola, 2012, Kim et al., 2011, Zhong et al., 2014, Egarter et al., 2015, Guo et al., 2015], resulting in state-of-the-art performance [Kolter and Jaakkola, 2012]. These works usually model the individual appliances by independent hidden Markov models (HMMs), which leads to a factorial HMM (FHMM) model describing the total consumption.

FHMMs, introduced by Ghahramani and Jordan [1997], are powerful tools for modeling times series generated from multiple independent sources, and are great for modeling speech with multiple people simultaneously talking [Rennie et al., 2009], or energy monitoring which we consider here [Kim et al., 2011]. Doing exact inference in FHMMs is NP hard; therefore, computationally efficient approximate methods have been the subject of study. Classic approaches include sampling methods, such as MCMC or particle filtering [Koller and Friedman, 2009] and variational Bayes methods [Wainwright and Jordan, 2007, Ghahramani and Jordan, 1997]. In practice, both methods are nontrivial to make work and we are not aware of any works that would have demonstrated good results in our application domain with the type of FHMMs we need to work and at practical scales.

In this paper we follow the work of Kolter and Jaakkola [2012] to model the NILM problem by FHMMs. The distinguishing features of FHMMs in this setting are that (i) the output is the sum of the output of the underlying HMMs (perhaps with some noise), and (ii) the number of transitions are small in comparison to the signal length. FHMMs with the first property are called *additive*. In this paper we derive an efficient, convex relaxation based method for FHMMs of the above type, which significantly outperforms the state-of-the-art algorithms. Our approach is based on revisiting relaxations to the integer programming formulation of Kolter and Jaakkola [2012]. In particular, we replace the quadratic programming relaxation of Kolter and Jaakkola, 2012 with a relaxation to an semi-definite program (SDP), which, based on the literature of relaxations is expected to be tighter and thus better. While SDPs are convex and could in theory be solved using interior-point (IP) methods in polynomial time [Malick et al., 2009], IP scales poorly with the size of the problem and is thus unsuitable to our large scale problem which may involve as many a million variables. To address this problem, capitalizing on the structure of our relaxation coming from our FHMM model, we develop a novel variant of ADMM [Boyd et al., 2011] that uses *Moreau-Yosida* regularization and combine it with a version of randomized rounding that is inspired by the the recent work of Park and Boyd [2015]. Experiments on synthetic and real data confirm that our method significantly outperforms other algorithms from the literature, and we expect that it may find its applications in other FHMM inference problems, too.

## 1.1 Notation

Throughout the paper, we use the following notation: $\mathbb{R}$ denotes the set of real numbers, $\mathbb{S}_+^n$ denotes the set of $n \times n$ positive semidefinite matrices, $\mathbb{I}_{\{E\}}$ denotes the indicator function of an event $E$ (that is, it is 1 if the event is true and zero otherwise), $\mathbf{1}$ denotes a vector of appropriate dimension whose entries are all 1. For an integer $K$, $[K]$ denotes the set $\{1, 2, \ldots, K\}$. $\mathcal{N}(\mu, \Sigma)$ denotes the Gaussian distribution with mean $\mu$ and covariance matrix $\Sigma$. For a matrix $A$, **trace**$(A)$ denotes its trace and **diag**$(A)$ denotes the vector formed by the diagonal entries of $A$.

## 2 System Model

Following Kolter and Jaakkola [2012], the energy usage of the household is modeled using an additive factorial HMM [Ghahramani and Jordan, 1997]. Suppose there are $M$ appliances in a household. Each of them is modeled via an HMM: let $P_i \in \mathbb{R}^{K_i \times K_i}$ denote the transition-probability matrix of appliance $i \in [M]$, and assume that for each state $s \in [K_i]$, the energy consumption of the appliance is constant $\mu_{i,s}$ ($\mu_i$ denotes the corresponding $K_i$-dimensional column vector $(\mu_{i,1}, \ldots, \mu_{i,K_i})^\top$). Denoting by $x_{t,i} \in \{0,1\}^{K_i}$ the indicator vector of the state $s_{t,i}$ of appliance $i$ at time $t$ (i.e., $x_{t,i,s} = \mathbb{I}_{\{s_{t,i}=s\}}$), the total power consumption at time $t$ is $\sum_{i\in[M]} \mu_i^\top x_{t,i}$, which we assume is observed with some additive zero mean Gaussian noise of variance $\sigma^2$: $y_t \sim \mathcal{N}(\sum_{i\in[M]} \mu_i^\top x_{t,i}, \sigma^2)$.[1]

Given this model, the maximum likelihood estimate of the appliance state vector sequence can be obtained by minimizing the log-posterior function

$$\arg\min_{x_{t,i}} \quad \sum_{t=1}^{T} \frac{(y_t - \sum_{i=1}^{M} x_{t,i}^\top \mu_i)^2}{2\sigma^2} - \sum_{t=1}^{T-1}\sum_{i=1}^{M} x_{t,i}^\top (\log P_i) x_{t+1,i} \qquad (1)$$

$$\text{subject to} \quad x_{t,i} \in \{0,1\}^{K_i}, \ \mathbf{1}^\top x_{t,i} = 1, \ i \in [M] \text{ and } t \in [T],$$

where $\log P_i$ denotes a matrix obtained from $P_i$ by taking the logarithm of each entry.

In our particular application, in addition to the signal's temporal structure, large changes in total power (in comparison to signal noise) contain valuable information that can be used to further improve the inference results (in fact, solely this information was used for energy disaggregation, e.g., by Dong et al., 2012, 2013, Figueiredo et al., 2012). This observation was used by Kolter and Jaakkola [2012] to amend the posterior with a term that tries to match the large signal changes to the possible changes in the power level when only the state of a single appliance changes.

Formally, let $\Delta y_t = y_{t+1} - y_t$, $\Delta\mu_{m,k}^{(i)} = \mu_{i,k} - \mu_{i,m}$, and define the matrices $E_{t,i} \in \mathbb{R}^{K_i \times K_i}$ by $(E_{t,i})_{m,k} = (\Delta y_t - \Delta\mu_{m,k}^{(i)})^2/(2\sigma_{\text{diff}}^2)$, for some constant $\sigma_{\text{diff}} > 0$. Intuitively, $(E_{t,i})_{m,k}$ is the negative log-likelihood (up to a constant) of observing a change $\Delta y_t$ in the power level when appliance $i$ transitions from state $m$ to state $k$ under some zero-mean Gaussian noise with variance $\sigma_{\text{diff}}^2$. Making the heuristic approximation that the observation noise and this noise are independent (which clearly does not hold under the previous model), Kolter and Jaakkola [2012] added the term $(-\sum_{t=1}^{T-1} \sum_{i=1}^{M} x_{t,i}^\top E_{t,i} x_{t+1,i})$ to the objective of (1), arriving at

$$\arg\min_{x_{t,i}} \quad f(x_1, \ldots, x_T) := \sum_{t=1}^{T} \frac{(y_t - \sum_{i=1}^{M} x_{t,i}^\top \mu_i)^2}{2\sigma^2} - \sum_{t=1}^{T-1} \sum_{i=1}^{M} x_{t,i}^\top (E_{t,i} + \log P_i) x_{t+1,i} \quad (2)$$

$$\text{subject to} \quad x_{t,i} \in \{0,1\}^{K_i}, \ \mathbf{1}^\top x_{t,i} = 1, \ i \in [M] \text{ and } t \in [T].$$

In the rest of the paper we derive an efficient approximate solution to (2), and demonstrate that it is superior to the approximate solution derived by Kolter and Jaakkola [2012] with respect to several measures quantifying the accuracy of load disaggregation solutions.

# 3 SDP Relaxation and Randomized Rounding

There are two major challenges to solve the optimization problem (2) exactly: (i) the optimization is over binary vectors $x_{t,i}$; and (ii) the objective function $f$, even when considering its extension to a convex domain, is in general non-convex (due to the second term). As a remedy we will relax (2) to make it an integer quadratic programming problem, then apply an SDP relaxation and randomized rounding to solve approximately the relaxed problem. We start with reviewing the latter methods.

## 3.1 Approximate Solutions for Integer Quadratic Programming

In this section we consider approximate solutions to the integer quadratic programming problem

$$\begin{aligned}
\text{minimize} \quad & f(x) = x^\top D x + 2d^\top x \\
\text{subject to} \quad & x \in \{0,1\}^n,
\end{aligned} \quad (3)$$

where $D \in \mathbb{S}_+^n$ is positive semidefinite, and $d \in \mathbb{R}^n$. While an exact solution of (3) can be found by enumerating all possible combination of binary values within a properly chosen box or ellipsoid, the running time of such exact methods is nearly exponential in the number $n$ of binary variables, making these methods unfit for large scale problems.

One way to avoid exponential running times is to replace (3) with a convex problem with the hope that the solutions of the convex problems can serve as a good starting point to find high-quality solutions to (3). The standard approach to this is to *linearize* (3) by introducing a new variable $X \in \mathbb{S}_+^n$ tied to $x$ trough $X = xx^\top$, so that $x^\top D x = \mathbf{trace}(DX)$, and then relax the nonconvex constraints $X = xx^\top$, $x \in \{0,1\}^n$ to $X \succeq xx^\top$, $\mathbf{diag}(X) = x$, $x \in [0,1]^n$. This leads to the relaxed SDP problem

$$\begin{aligned}
\text{minimize} \quad & \mathbf{trace}(D^\top X) + 2d^\top x \\
\text{subject to} \quad & \begin{bmatrix} 1 & x^\top \\ x & X \end{bmatrix} \succeq 0, \quad \mathbf{diag}(X) = x, \quad x \in [0,1]^n
\end{aligned} \quad (4)$$

By introducing $\hat{X} = \begin{bmatrix} 1 & x^\top \\ x & X \end{bmatrix}$ this can be written in the compact SDP form

$$
\begin{aligned}
\text{minimize} \quad & \mathbf{trace}(\hat{D}^\top \hat{X}) \\
\text{subject to} \quad & \hat{X} \succeq 0, \quad \mathcal{A}\hat{X} = b.
\end{aligned}
\tag{5}
$$

where $\hat{D} = \begin{bmatrix} 0 & d^\top \\ d & D \end{bmatrix} \in \mathbb{S}_+^{n+1}$, $b \in \mathbb{R}^m$ and $\mathcal{A} : \mathbb{S}_+^n \to \mathbb{R}^m$ is an appropriate linear operator. This general SDP optimization problem can be solved with arbitrary precision in polynomial time using interior-point methods [Malick et al., 2009, Wen et al., 2010]. As discussed before, this approach becomes impractical in terms of both the running time and the required memory if either the number of variables or the optimization constraints are large [Wen et al., 2010]. We will return to the issue of building scaleable solvers for NILM in Section 5.

Note that introducing the new variable $X$, the problem is projected into a higher dimensional space, which is computationally more challenging than just simply relaxing the integrality constraint in (3), but leads to a tighter approximation of the optimum (c.f., Park and Boyd, 2015; see also Lovász and Schrijver, 1991, Burer and Vandenbussche, 2006).

To obtain a feasible point of (3) from the solution of (5), we still need to change the solution $x$ to a binary vector. This can be done via randomized rounding [Park and Boyd, 2015, Goemans and Williamson, 1995]: Instead of letting $x \in [0,1]^n$, the integrality constraint $x \in \{0,1\}^n$ in (3) can be replaced by the inequalities $x_i(x_i - 1) \geq 0$ for all $i \in [n]$. Although these constraints are nonconvex, they admit an interesting probabilistic interpretation: the optimization problem

$$
\begin{aligned}
\text{minimize} \quad & \mathbb{E}_{w \sim \mathcal{N}(\mu, \Sigma)}[w^\top D w + 2 d^\top w] \\
\text{subject to} \quad & \mathbb{E}_{w \sim \mathcal{N}(\mu, \Sigma)}[w_i(w_i - 1)] \geq 0, \qquad i \in [n], \quad \mu \in \mathbb{R}^n, \quad \Sigma \succeq 0
\end{aligned}
$$

is equivalent to

$$
\begin{aligned}
\text{minimize} \quad & \mathbf{trace}((\Sigma + \mu\mu^\top)D) + 2 d^\top \mu \\
\text{subject to} \quad & \Sigma_{i,i} + \mu_i^2 - \mu_i \geq 0, \qquad i \in [n],
\end{aligned}
\tag{6}
$$

which is in the form of (4) with $X = \Sigma + \mu\mu^\top$ and $x = \mu$ (above, $\mathbb{E}_{x \sim P}[f(x)]$ stands for $\int f(x) dP(x)$). This leads to the rounding procedure: starting from a solution $(x^*, X^*)$ of (4), we randomly draw several samples $w^{(j)}$ from $\mathcal{N}(x^*, X^* - x^* x^{*\top})$, round $w_i^{(j)}$ to 0 or 1 to obtain $x^{(j)}$, and keep the $x^{(j)}$ with the smallest objective value. In a series of experiments, Park and Boyd [2015] found this procedure to be better than just naively rounding the coordinates of $x^*$.

## 4 An Efficient Algorithm for Inference in FHMMs

To arrive at our method we apply the results of the previous subsection to (2). To do so, as mentioned at the beginning of the section, we need to change the problem to a convex one, since the elements of the second term in the objective of (2), $-x_{t,i}^\top (E_{t,i} + \log P_i) x_{t+1,i}$ are not convex. To address this issue, we relax the problem by introducing new variables $Z_{t,i} = x_{t,i} x_{t+1,i}^\top$ and replace the constraint $Z_{t,i} = x_{t,i} x_{t+1,i}^\top$ with two new ones:

$$
Z_{t,i}\mathbf{1} = x_{t,i} \quad \text{and} \quad Z_{t,i}^\top \mathbf{1} = x_{t+1,i}.
$$

To simplify the presentation, we will assume that $K_i = K$ for all $i \in [M]$. Then problem (2) becomes

$$
\begin{aligned}
\arg\min_{x_{t,i}} \quad & \sum_{t=1}^{T} \left\{ \frac{1}{2\sigma^2} \left(y_t - x_t^\top \mu\right)^2 - p_t^\top z_t \right\} \\
\text{subject to} \quad & x_t \in \{0,1\}^{MK}, \qquad t \in [T], \\
& \hat{z}_t \in \{0,1\}^{MKK}, \qquad t \in [T-1], \\
& \mathbf{1}^\top x_{t,i} = 1, \qquad t \in [T] \text{ and } i \in [M], \\
& Z_{t,i}\mathbf{1}^\top = x_{t,i}, \qquad Z_{t,i}^\top \mathbf{1}^\top = x_{t+1,i}, \qquad t \in [T-1] \text{ and } i \in [M],
\end{aligned}
\tag{7}
$$

---

**Algorithm 1** ADMM-RR: Randomized rounding algorithm for suboptimal solution to (2)

---
**Given:** number of iterations: itermax, length of input data: $T$
Solve the optimization problem (8): Run Algorithm 2 to get $X_t^*$ and $z_t^*$
Set $x_t^{best} := z_t^*$ and $X_t^{best} := X_t^*$ for $t = 1, \ldots, T$
**for** $t = 2, \ldots, T - 1$ **do**
   Set $x := [x_{t-1}^{best\top}, x_t^{best\top}, x_{t+1}^{best\top}]^\top$
   Set $X := \mathbf{block}(X_{t-1}^{best}, X_t^{best}, X_{t+1}^{best})$ where $\mathbf{block}(\cdot, \cdot)$ constructs block diagonal matrix from input arguments
   Set $f^{best} := \infty$
   Form the covariance matrix $\Sigma := X - xx^T$ and find its Cholesky factorization $LL^\top = \Sigma$.
   **for** $k = 1, 2, \ldots,$ itermax **do**
      Random sampling: $z^k := x + Lw$, where $w \sim \mathcal{N}(0, I)$
      Round $z^k$ to the nearest integer point $x^k$ that satisfies the constraints of (7)
      If $f^{best} > f_t(x^k)$ then update $x_t^{best}$ and $X_t^{best}$ from the corresponding entries of $x^k$ and $x^k x^{k\top}$, respectively
   **end for**
**end for**

---

where $x_t^\top = [x_{t,1}^\top, \ldots, x_{t,M}^\top]$, $\mu^\top = [\mu_1^\top, \ldots, \mu_M^\top]$, $z_t^\top = [\mathbf{vec}(Z_{t,1})^\top, \ldots, \mathbf{vec}(Z_{t,M})^\top]$ and $p_t^\top = [\mathbf{vec}(E_{t,1} + \log P_1), \ldots, \mathbf{vec}(\log P_T)]$, with $\mathbf{vec}(A)$ denoting the column vector obtained by concatenating the columns of $A$ for a matrix $A$. Expanding the first term of (7) and following the relaxation method of Section 3.1, we get the following SDP problem:[2]

$$
\begin{aligned}
\arg \min_{X_t, z_t} \quad & \sum_{t=1}^{T} \mathbf{trace}(D_t^\top X_t) + d_t^\top z_t \\
\text{subject to} \quad & \mathcal{A}X_t = b, \qquad \mathcal{B}X_t + \mathcal{C}z_t + \mathcal{E}X_{t+1} = g, \\
& X_t \succeq 0, \qquad X_t, z_t \geq 0.
\end{aligned}
\tag{8}
$$

Here $\mathcal{A} : \mathbb{S}_+^{MK+1} \to \mathbb{R}^m$, $\mathcal{B}, \mathcal{E} : \mathbb{S}_+^{MK+1} \to \mathbb{R}^{m'}$ and $\mathcal{C} \in \mathbb{R}^{MKK \times m'}$ are all appropriate linear operators, and the integers $m$ and $m'$ are determined by the number of equality constraints, while $D_t = \frac{1}{2\sigma^2} \begin{bmatrix} 0 & -y_t \mu^\top \\ -y_t \mu & \mu \mu^\top \end{bmatrix}$ and $d_t = p_t$. Notice that (8) is a simple, though huge-dimensional SDP problem in the form of (5) where $\hat{D}$ has a special block structure.

Next we apply the randomized rounding method from Section 3.1 to provide an approximate solution to our original problem (2). Starting from an optimal solution $(z^*, X^*)$ of (8), and utilizing that we have an SDP problem for each time step $t$, we obtain Algorithm 1 that performs the rounding sequentially for $t = 1, 2, \ldots, T$. However we run the randomized method for three consecutive time steps, since $X_t$ appears at both time steps $t - 1$ and $t + 1$ in addition to time $t$ (cf., equation 9). Following Park and Boyd [2015], in the experiments we introduce a simple greedy search within Algorithm 1: after finding the initial point $x^k$, we greedily try to objective the target value by change the status of a single appliance at a single time instant. The search stops when no such improvement is possible, and we use the resulting point as the estimate.

## 5 ADMM Solver for Large-Scale, Sparse Block-Structured SDP Problems

Given the relaxation and randomized rounding presented in the previous subsection all that remains is to find $X_t^*, z_t^*$ to initialize Algorithm 1. Although interior point methods can solve SDP problems efficiently, even for problems with sparse constraints as (4), the running time to obtain an $\epsilon$ optimal solution is of the order of $n^{3.5} \log(1/\epsilon)$ [Nesterov, 2004, Section 4.3.3], which becomes prohibitive in our case since the number of variables scales linearly with the time horizon $T$.

As an alternative solution, *first-order* methods can be used for large scale problems [Wen et al., 2010]. Since our problem (8) is an SDP problem where the objective function is separable, ADMM is a promising candidate to find a near-optimal solution. To apply ADMM, we use the *Moreau-Yosida* quadratic regularization [Malick et al., 2009], which is well suited for the primal formulation we

**Algorithm 2** ADMM for sparse SDPs of the form (8)

---

**Given:** length of input data: $T$, number of iterations: itermax.
Set the initial values to zero. $W_t^0, P_t^0, S^0 = \mathbf{0}, \lambda_t^0 = \mathbf{0}, \nu_t^0 = \mathbf{0}$, and $r_t^0, h_t^0 = \mathbf{0}$
Set $\mu = 0.001$ {Default step-size value}
**for** $k = 0, 1, \ldots,$ itermax **do**
    **for** $t = 1, 2, \ldots, T$ **do**
        Update $P_t^k, W_t^k, \lambda^k, S_t^k, r_t^k, h_t^k$, and $\nu_t^k$, respectively, according to (11) (Appendix A).
    **end for**
**end for**

---

consider. When implementing ADMM over the variables $(X_t, z_t)_t$, the sparse structure of our constraints allows to consider the SDP problems for each time step $t$ sequentially:

$$
\begin{aligned}
\arg\min_{X_t, z_t} \quad & \mathbf{trace}(D_t^\top X_t) + d_t^\top z_t \\
\text{subject to} \quad & \mathcal{A}X_t = b, \\
& \mathcal{B}X_t + \mathcal{C}z_t + \mathcal{E}X_{t+1} = g, \\
& \mathcal{B}X_{t-1} + \mathcal{C}z_{t-1} + \mathcal{E}X_t = g, \\
& X_t \succeq 0, \qquad X_t, z_t \geq 0 \,.
\end{aligned}
\tag{9}
$$

The regularized Lagrangian function for (9) is[3]

$$
\begin{aligned}
\mathcal{L}_\mu = & \mathbf{trace}(D^\top X) + d^\top z + \frac{1}{2\mu}\|X - S\|_F^2 + \frac{1}{2\mu}\|z - r\|_2^2 + \lambda^\top(b - \mathcal{A}X) \\
& + \nu^\top(g - \mathcal{B}X - \mathcal{C}z - \mathcal{E}X_+) + \nu_-^\top(g - \mathcal{B}X_- - \mathcal{C}z_- - \mathcal{E}X) \\
& - \mathbf{trace}(W^\top X) - \mathbf{trace}(P^\top X) - h^\top z,
\end{aligned}
\tag{10}
$$

where $\lambda, \nu, W \geq 0$, $P \succeq 0$, and $h \geq 0$ are dual variables, and $\mu > 0$ is a constant. By taking the derivatives of $\mathcal{L}_\mu$ and computing the optimal values of $X$ and $z$, one can derive the standard ADMM updates, which, due to space constraints, are given in Appendix A. The final algorithm, which updates the variables for each $t$ sequentially, is given by Algorithm 2.

Algorithms 1 and 2 together give an efficient algorithm for finding an approximate solution to (2) and thus also to the inference problem of additive FHMMs.

## 6 Learning the Model

The previous section provided an algorithm to solve the inference part of our energy disaggregation problem. However, to be able to run the inference method, we need to set up the model. To learn the HMMs describing each appliance, we use the method of Kontorovich et al. [2013] to learn the transition matrix, and the spectral learning method of Anandkumar et al. [2012] (following Mattfeld, 2014) to determine the emission parameters.

However, when it comes to the specific application of NILM, the problem of *unknown, time-varying bias* also needs to be addressed, which appears due to the presence of unknown/unmodeled appliances in the measured signal. A simple idea, which is also followed by Kolter and Jaakkola [2012], is to use a "generic model" whose contribution to the objective function is downweighted. Surprisingly, incorporating this idea in the FHMM inference creates some unexpected challenges.[4]

Therefore, in this work we come up with a *practical, heuristic* solution tailored to NILM. First we identify all electric events defined by a large change $\Delta y_t$ in the power usage (using some ad-hoc threshold). Then we discard all events that are similar to any possible level change $\Delta\mu_{m,k}^{(i)}$. The remaining large jumps are regarded as coming from a generic HMM model describing the unregistered appliances: they are clustered into $K - 1$ clusters, and an HMM model is built where each cluster is regarded as power usage coming from a single state of the unregistered appliances. We also allow an "off state" with power usage 0.

# 7 Experimental Results

We evaluate the performance of our algorithm in two setups:[5] we use a synthetic dataset to test the inference method in a controlled environment, while we used the REDD dataset of Kolter and Johnson [2011] to see how the method performs on non-simulated, "real" data. The performance of our algorithm is compared to the structured variational inference (SVI) method of Ghahramani and Jordan [1997], the method of Kolter and Jaakkola [2012] and that of Zhong et al. [2014]; we shall refer to the last two algorithms as KJ and ZGS, respectively.

## 7.1 Experimental Results: Synthetic Data

The synthetic dataset was generated randomly (the exact procedure is described in Appendix C). To evaluate the performance, we use *normalized disaggregation error* as suggested by Kolter and Jaakkola [2012] and also adopted by Zhong et al. [2014]. This measures the reconstruction error for each individual appliance. Given the true output $y_{t,i}$ and the estimated output $\hat{y}_{t,i}$ (i.e. $\hat{y}_{t,i} = \mu_i^\top \hat{x}_{t,i}$), the error measure is defined as

$$\text{NDE} = \sqrt{\sum_{t,i}(y_{t,i} - \hat{y}_{t,i})^2 / \sum_{t,i}(y_{t,i})^2}.$$

Figures 1 and 2 show the performance of the algorithms as the number HMMs ($M$) (resp., number of states, $K$) is varied. Each plot is a report for $T = 1000$ steps averaged over 100 random models and realizations, showing the mean and standard deviation of NDE. Our method, shown under the label ADMM-RR, runs ADMM for 2500 iterations, runs the local search at the end of each 250 iterations, and chooses the result that has the maximum likelihood. ADMM is the algorithm which applies naive rounding. It can be observed that the variational inference method is significantly outperformed by all other methods, while our algorithm consistently obtained better results than its competitors, KJ coming second and ZGS third.

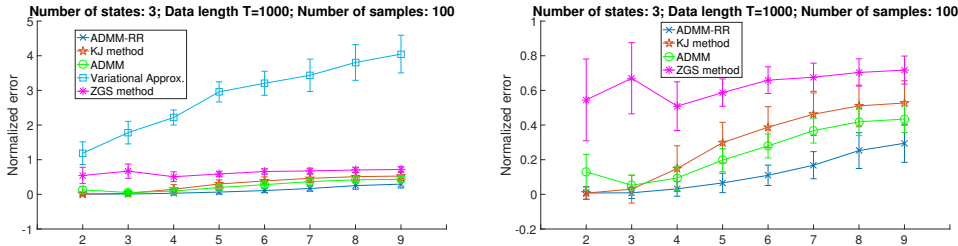

Figure 1: Disaggregation error varying the number of HMMs.

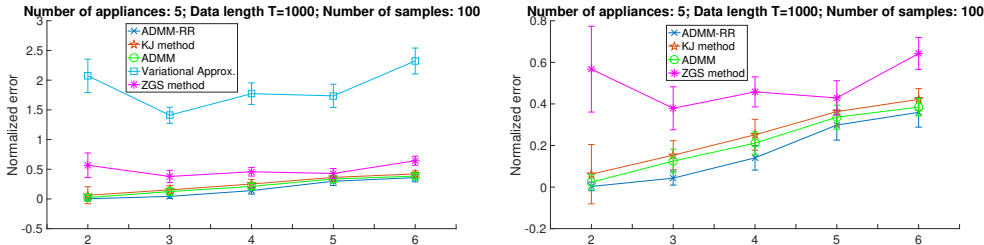

Figure 2: Disaggregation error varying the number of states.

## 7.2 Experimental Results: Real Data

In this section, we also compared the 3 best methods on the real dataset REDD [Kolter and Johnson, 2011]. We use the first half of the data for training and the second half for testing. Each HMM (i.e.,

| Appliance | ADMM-RR | KJ method | ZGS method |
|---|---|---|---|
| 1 Oven-3 | **61.70/78.30%** | 27.62/72.32% | 5.35/15.04% |
| 2 Fridge | **90.22/97.63%** | 41.20/97.46% | 46.89/87.10% |
| 3 Microwave | 12.40/74.74% | **13.40/96.32%** | 4.55/45.07% |
| 4 Bath. GFI-12 | **50.88/60.25%** | 12.87/51.46% | 6.16/42.67% |
| 5 Kitch. Out.-15 | **69.23/98.85%** | 16.66/79.47% | 5.69/26.72% |
| 6 Wash./Dry.-20-A | 98.23/93.80% | 70.41/98.19% | 15.91/35.51% |
| 7 Unregistered-A | 94.27/87.80% | 85.35/25.91% | 57.43/99.31% |
| 8 Oven-4 | 25.41/76.37% | 13.60/78.59% | 9.52/12.05% |
| 9 Dishwasher-6 | 54.53/90.91% | 25.20/98.72% | 29.42/31.01% |
| 10 Wash./Dryer-10 | **21.92/63.58%** | 18.63/25.79% | 7.79/3.01% |
| 11 Kitch. Out.-16 | 17.88/79.04% | 8.87/100% | 0.00/0.00% |
| 12 Wash./Dry.-20-B | 98.19/28.31% | 72.13/77.10% | 27.44/71.25% |
| 13 Unregistered-B | 97.78/91.73% | 96.92/73.97% | 33.63/99.98% |
| Average | **60.97/78.56%** | 38.68/75.02% | 17.97/36.22% |

Table 1: Comparing the disaggregation performance of three different algorithms: precision/recall. Bold numbers represent statistically better performance on both measures.

appliance) is trained separately using the associated circuit level data, and the HMM corresponding to unregistered appliances is trained using the main panel data. In this set of experiments we monitor appliances consuming more than 100 watts. ADMM-RR is run for 1000 iterations, and the local search is run at the end of each 250 iterations, and the result with the largest likelihood is chosen. To be able to use the ZGS method on this data, we need to have some prior information about the usage of each appliance; the authors suggestion is to us national energy surveys, but in the lack of this information (also about the number of residents, type of houses, etc.) we used the training data to extract this prior knowledge, which is expected to help this method.

Detailed results about the precision and recall of estimating which appliances are 'on' at any given time are given in Table 1. In Appendix D we also report the error of the total power usage assigned to different appliances (Table 2), as well as the amount of assigned power to each appliance as a percentage of total power (Figure 3). As a summary, we can see that our method consistently outperformed the others, achieving an average precision and recall of $60.97\%$ and $78.56\%$, with about $50\%$ better precision than KJ with essentially the same recall ($38.68/75.02\%$), while significantly improving upon ZGS ($17.97/36.22\%$). Considering the error in assigning the power consumption to different appliances, our method achieved about $30 - 35\%$ smaller error (ADMM-RR: $2.87\%$, KJ: $4.44\%$, ZGS: $3.94\%$) than its competitors.

In our real-data experiments, there are about 1 million decision variables: $M = 7$ or $6$ appliances (for phase A and B power, respectively) with $K = 4$ states each and for about $T = 30,000$ time steps for one day, 1 sample every 6 seconds. KJ and ZGS solve quadratic programs, increasing their memory usage (14GB vs 6GB in our case). On the other hand, our implementation of their method, using the commercial solver MOSEK inside the Matlab-based YALMIP [Löfberg, 2004], runs in 5 minutes, while our algorithm, which is purely Matlab-based takes 5 hours to finish. We expect that an optimized C++ version of our method could achieve a significant speed-up compared to our current implementation.

# 8    Conclusion

FHMMs are widely used in energy disaggregation. However, the resulting model has a huge (factored) state space, making standard inference FHMM algorithms infeasible even for only a handful of appliances. In this paper we developed a scalable approximate inference algorithm, based on a semidefinite relaxation combined with randomized rounding, which significantly outperformed the state of the art in our experiments. A crucial component of our solution is a scalable ADMM method that utilizes the special block-diagonal-like structure of the SDP relaxation and provides a good initialization for randomized rounding. We expect that our method may prove useful in solving other FHMM inference problems, as well as in large scale integer quadratic programming.

# Acknowledgements

This work was supported in part by the Alberta Innovates Technology Futures through the Alberta Ingenuity Centre for Machine Learning and by NSERC. K. is indebted to Pooria Joulani and Mohammad Ajallooeian, whom provided much useful technical advise, while all authors are grateful for Zico Kolter for sharing his code.

## Footnotes

[1]Alternatively, we can assume that the power consumption $y_{t,i}$ of each appliance is normally distributed with mean $\mu_i^\top x_{t,i}$ and variance $\sigma_i^2$, where $\sigma^2 = \sum_{i\in[M]} \sigma_i^2$, and $y_t = \sum_{i\in[M]} y_{t,i}$.

[2]The only modification is that we need to keep the equality constraints in (7) that are missing from (3).

[3]We drop the subscript $t$ and replace $t + 1$ and $t - 1$ with $+$ and $-$ signs, respectively.

[4]For example, the incorporation of this generic model breaks the derivation of the algorithm of Kolter and Jaakkola [2012]. See Appendix B for a discussion of this.

[5]Our code is available online at `https://github.com/kiarashshaloudegi/FHMM_inference`.

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
