[Supplementary Material · SDP-FHMM.pdf]

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

## A  ADMM updates

In this section we derive the ADMM updates for the regularized Lagrangian $\mathcal{L}_\mu$ given by (10). Taking derivatives with respect to $X$ and $z$ and setting them to zeros, we get

$$\nabla_X \mathcal{L}_\mu = D + \frac{1}{\mu}(X - S) - \mathcal{A}^\top \lambda - \mathcal{B}^\top \nu - \mathcal{E}^\top \nu_- - W - P = 0,$$

$$X^* = S + \mu(\mathcal{A}^\top \lambda + \mathcal{B}^\top \nu + \mathcal{E}^\top \nu_- + W + P - D)$$

and

$$\nabla_z \mathcal{L}_\mu = d + \frac{1}{\mu}(z - r) - \mathcal{C}^\top \nu - h = 0,$$

$$z^* = r + \mu(\mathcal{C}^\top \nu + h - d).$$

Substituting $X = X^*$ and $z = z^*$ in (10) defines $\hat{\mathcal{L}}_\mu$. Then the standard ADMM iteration yields

$$P^{k+1} = \arg\min_{P \succeq 0} \hat{\mathcal{L}}_\mu(S^k, P, W^k, \lambda^k, \nu^k, \nu_-^k),$$

$$W^{k+1} = \arg\min_{W \geq 0} \hat{\mathcal{L}}_\mu(S^k, P^{k+1}, W, \lambda^k, \nu^k, \nu_-^k),$$

$$\lambda^{k+1} = \arg\min_{\lambda} \hat{\mathcal{L}}_\mu(S^k, P^{k+1}, W^{k+1}, \lambda, \nu^k, \nu_-^k),$$

$$S^{k+1} = S^k + \mu(\mathcal{A}^\top \lambda^{k+1} + \mathcal{B}^\top \nu^k + \mathcal{E}^\top \nu_-^{k+1} + W^{k+1} + P^{k+1} - D),$$

$$r^{k+1} = r^k + \mu(\mathcal{C}^\top \nu^k + h^k - d),$$

$$h^{k+1} = \arg\min_{h \geq 0} \hat{\mathcal{L}}_\mu(r^{k+1}, \nu^k),$$

$$\nu^{k+1} = \arg\min_{\nu} \hat{\mathcal{L}}_\mu(S^{k+1}, P^{k+1}, W^{k+1}, \lambda^{k+1}, \nu, \nu_-^{k+1}, h^{k+1}, r^{k+1}).$$

By rearranging the terms in $\hat{\mathcal{L}}_\mu$, the following update equations can be found:

$$W^{k+1} = \max\{(D - \mathcal{A}^\top \lambda^k - \mathcal{B}^\top \nu^k - \mathcal{E}^\top \nu_-^k - P^k - D - S^k/\mu), \mathbf{0}\},$$

$$P^{k+1} = (D - \mathcal{A}^\top \lambda^k - \mathcal{B}^\top \nu^k - \mathcal{E}^\top \nu_-^k - W^k - D - S^k/\mu)_+,$$

$$\lambda^{k+1} = \frac{1}{\mu}(\mathcal{A}\mathcal{A}^\top)^\dagger \big(b - \mathcal{A}(\mathcal{B}^\top \nu^k + \mathcal{E}^\top \nu_-^k + W^{k+1} + P^{k+1} - D)\big),$$

$$h^{k+1} = \max\{d - \mathcal{C}^\top \nu^k - r^k/\mu, \mathbf{0}\},$$

$$\nu^{k+1} = \frac{1}{\mu}(\mathcal{B}\mathcal{B}^\top + \mathcal{C}\mathcal{C}^\top + \mathcal{E}\mathcal{E}^\top)^\dagger \big(g - \mathcal{B}(S^{k+1} + \mu(\mathcal{A}^\top \lambda^{k+1} + \mathcal{E}^\top \nu_-^{k+1} + W^{k+1} + P^{k+1} - D))$$
$$- \mathcal{C}(r^{k+1} + \mu(h^{k+1} - d)) - \mathcal{E}(S_+^k + \mu(\mathcal{A}^\top \lambda_+^k + \mathcal{B}^\top \nu_+^k + W_+^k + P_+^k - D))\big).$$
$$\tag{11}$$

Here $\max : \mathcal{X} \times \mathcal{X} \to \mathcal{X}$ works elementwise, and for any square matrix $A$, $A^\dagger$ denotes the *Moore-Penrose pseudo inverse*, and for any real symmetric matrix $A$, $A_+$ is the projection of $A$ onto the positive semidefinite cone (if the spectral decomposition of $A$ is given by $A = \sum_i \lambda_i v_i v_i^\top$, where $\lambda_i$ and $v_i$ are the $i^{\text{th}}$ eigenvalue and eigenvector of $A$, respectively, then $A_+ = \sum_{\lambda_i > 0} \lambda_i v_i v_i^\top$). Note that the projections are done on matrices of small size. Note also that the pseudo-inverses of the matrices involved need only be calculated once.

## B  Discussion of the Derivation in Kolter and Jaakkola [2012] in the Presence of the "Generic Model"

The "generic model" affects the derivation of the algorithm of Kolter and Jaakkola [2012] as follows. The authors of this paper claim to derive the final optimization problem given in equation (15) of their paper from (9) and (10) as follows: equation (9) defines the problem $\min_{z \in Z, Q \in A} f_1(z, Q)$, while (10) defines the problem $\min_{z' \in Z', Q \in A} f_2(z', Q)$ where $z' = g(z)$. Here, $z, z'$ are variables that describe the state of the "generic model" over time.

| Appliance | Actual power | ADMM-RR error | KJ error | ZGS error |
|---|---|---|---|---|
| 1 Oven | 2.26% | 0.82% | **0.08%** | 1.47% |
| 2 Fridge | 17.45% | 0.98% | 8.38% | **0.44%** |
| 3 Micro. | 4.79% | **0.49%** | 1.89% | 4.01% |
| 4 Bath. GFI | 2.10% | **0.13%** | 0.14% | 0.17% |
| 5 Kitch. Out. | 1.77% | 0.41% | 0.85% | **0.13%** |
| 6 Wash./Dry. | 13.54% | **0.36%** | 0.61% | 7.20% |
| 7 Unregistered | 14.45% | **1.46%** | 9.64% | 16.87% |
| 8 Oven | 3.14% | 8.09% | 8.49% | **1.44%** |
| 9 Dishwasher | 8.07% | 3.18% | 8.24% | **0.60%** |
| 10 Wash./Dryer | 1.96% | 4.02% | **0.54%** | 1.48% |
| 11 Kitch. Out. | 0.27% | 0.96% | 1.48% | **0.71%** |
| 12 Wash./Dry. | 13.54% | 9.47% | 8.02% | **7.20%** |
| 13 Unregistered | 16.17% | **6.92%** | 9.36% | 9.51% |
| Total | 100% | – | – | – |
| Average | – | **2.87%** | 4.44% | 3.94% |
| Std dev. | – | **3.26%** | 4.14% | 5.00% |
| Median | – | **0.98%** | 1.89% | 1.47% |

Table 2: Energy disaggregation error as a percentage of total energy for three different algorithms.

The claim in the paper is that with some set $B$ (coming from their "one-at-a-time" constraint), $\min_{z \in Z, z' \in Z', Q \in A \cap B, z' = g(z)} f_1(z, A) + f_2(z', Q)$ is equivalent to the minimization problem in equation (15). However, careful checking the derivation shows that (15) is equivalent to $\min_{z \in Z, z' \in Z', Q \in A \cap B} f_1(z, A) + \min_{z' \in Z'} f_2(z', Q)$, which is smaller in general.

## C   Generating the Synthetic Dataset

The synthetic dataset used in the experiments was generated in the following way: The power levels corresponding to each on state ($\mu$) were generated uniformly at random from $[100, 4500]$ with the additional constraint that the difference of any two non-zero levels must be greater than 100 (to encourage identifiability). The levels for "off states" were set to 0. The transition matrices for each appliance were generated the following way: diagonal elements for "off states" were drawn uniformly at random from $[0, 35]$ and for on-states from $[0, 30]$, while non-diagonal elements were selected from $[0, 1]$ to ensure sparse transitions. Finally, the data matrices were normalized to ensure they are proper transition matrices. The output of each appliance was subject to an additive Gaussian noise with variance $\sigma \in [0, 6]$ selected proportionally to the energy consumption level of the given on state, and 1 for off states.

## D   Additional Results for the Real-Data Experiment

In Table 1 we provided prediction and recall values for our experiments on real data. As promised, here we provide some additional results about these experiments: Table 2 presents the total power usage assigned to different appliances, and Figure 3 shows the amount of assigned power to each appliance.

Figure 3: Total energy assigned to different appliances.