[Reviews · NeurIPS 2016]

Reviewer 1

Summary

The paper investigates the problem of estimating the power consumed by each appliances in a household from the time series measurements of total energy consumed. The authors solve the problem with an additive factorial HMM model and infer the estimates using a few clever tricks including casting it as a convex semidefinite relaxation, random rounding, and an efficient and scalable alternating direction method of multipliers (ADMM). They demonstrates its efficacy on a simulated data set.

Qualitative Assessment

This paper tackles an intriguing problem and is well-written. Their solution, which is complex and contains several components, is explained in stages where each stage naturally follows from the previous one. Even for a reader not familiar with the problem domain, the paper is self-contained. After reading the paper, the one question that was unclear is how are the number of appliances estimated? Perhaps, the authors could elaborate on this in Section 5. It would also be useful to know how sensitive this choice is during the empirical evaluation.

Confidence in this Review

2-Confident (read it all; understood it all reasonably well)


Reviewer 2

Summary

The paper proposes refined optimization methods for learning additive factorial HMMs, motivated by the application to energy disaggregation. The method is compared to baseline approaches both on synthetic and real-world data sets.

Qualitative Assessment

The technical quality seems very solid. Overall, the presentation is good, but the authors should do a check for grammatical errors. The potential societal impact of this work is high. I would rank the originality of lower, as there exists already quite a bit of work in this direction. The experiments suggest substantial improvements over existing methods, so I'd consider this a significant improvement over the state-of-the-art. In their rebuttal, the authors carefully addressed concerns by some of the reviewers about the novelty and scalability of their method, and the comparison with results obtained by Kolter and Jaakkola (2012).

Confidence in this Review

1-Less confident (might not have understood significant parts)


Reviewer 3

Summary

This paper presents an efficient approximate solution for the task of energy disaggregation for home appliance monitoring formulated as a binary quadratic program -- equation (2). First equation (2) is relaxed into a convex problem (7) by introducing new variables. Then, it is relaxed again as a SDP problem with continuous variables leading to problem (8). Also, they proposed to use a variant of alternating direction method of multipliers for solving this large SDP. Binary variable are then obtained via via randomized rounding. The proposed solution appear to be superior to the one previously proposed by Kolter & Jaakkola (2012) with respect to several measures quantifying the accuracy of load disaggregation solutions.

Qualitative Assessment

The paper is well-written, almost self contained, easy to read and represents a significant piece of work. Its global organization could be improved since we have a subsection 1.1 without 1.2, a 3.1 without 3.2 and a 4.1 without 4.2. The interesting point of the paper is the global methodology used to solve the initial optimization problem. The whole Machine learning community would benefit from the techniques used in this paper. details/typos L75: in the indicator function equation 1 and 2: K -> K_i L189: yet.Kolter -> yet. Kolter

Confidence in this Review

2-Confident (read it all; understood it all reasonably well)


Reviewer 4

Summary

This paper studies the task of energy disaggregation load minotoring. The authors formulated the problem as an integer quadratic programming optimization problem, and then apply standard SDP relaxation and randomized rounding.

Qualitative Assessment

The theory part of this paper is a bit incremental. My main concern is on the experimental study of this paper: (1) Since the real dataset (REDD) used in this paper is the same as that in Kolter&Johnson (KJ), I took a quick look at the experiments in KJ. It seems that the precision/recall in KJ is much higher (average 87.2%/60.3% over 7 appliances), even higher than the proposed algorithm of this paper (in Table 1). This looks a bit strange to me, or, one of the two results must have issues. (2) There is no comparison on the running time in the experimental study. I feel this comparison is necessary since, as mentioned in the introduction, the goal of this paper is to "develop a scalable, computationally efficient method" I actually don't know how large is the tested dataset, and thus I don't know why solving SDP directly is infeasible here. About related work (second paragraph of intro), there are quite a few paper mentioned in the introduction that are published after 2012 (e.g., Zhong et. al. 2014). It will help to explain why KJ is the state-of-the-art? Minors: -- Line 75, what is the meaning of I_{} s_{t,i} = s? -- Title of Sec. 6. Why say "Synthetic Data Set"?

Confidence in this Review

2-Confident (read it all; understood it all reasonably well)


Reviewer 5

Summary

The authors provide a method used to estimate the energy consumption in a domestic house. They propose a system based on an additive factorial hidden Markov model and, with respect to previous implementations, they add an additional constraint to the objective function to be minimized, in order to account sudden changes in the level of power consumption.

Qualitative Assessment

The paper is really interesting and it propose a technical sound methodologies, which can be applied to tackle the important issue of managing energy consumption. I was already positive for what concerns the value of novelty in the presented study, comments of reviewers and authors feedback confirmed my position. My most serious concern at this point is the same raised from some reviewers, about providing a study on the computational complexity of the method proposed. Indeed, the authors discuss it in the rebuttal and I think it should be included in the final version of the paper. However, given the nature of the problem tackled, I believe a more formal study should be provided. Finally, the paper is quite technical and I believe a poster presentation would be much more suitable to present it.

Confidence in this Review

2-Confident (read it all; understood it all reasonably well)


Reviewer 6

Summary

This paper proposed method to find better approximate solution for energy disaggregation or non-intrusive load monitoring (NILM) problem based on the previous work done by Kolter & Jaakkola (2012) who used an additive factorial HMM to model the energy consumption. The authors combined a SDP relaxation and randomized rounding (Park & Boyd, 2015) and applied ADMM (Boyd, 2010). The proposed method achieved superior performance to the existing method.

Qualitative Assessment

It is meaningful that the proposed method perform outstandingly well on simulation results in aspects of accuracy. However, this paper lacks academic novelty and originality in that the proposed method solely combined existing works. The authors argue that proposed algorithm is scalable and computationally efficient, but there is no explanation about computational complexity on method or experimental results. It is necessary to show effectiveness in aspects of computational complexity. Alternatively, comparison with existing methods on execution time or experiments using large-scale data could be provided. This paper is well written overall and states essential information. The author should modify some mistakes in expression including the title of section 6 and its subsection, line 75 and line 223. It would be better to represent important figures or tables on the manuscript not on the supplementary material in spite of the limited space. In addition, conclusion part is too short.

Confidence in this Review

2-Confident (read it all; understood it all reasonably well)